# Control and Trajectory Planning of an Autonomous Bicycle Robot

**Masiala Mavungu**

Faculty of Engineering and Built Environment, University of Johannesburg,
Johannesburg 2092, Auckland, South Africa; masialam@uj.ac.za

**Abstract:** This paper addresses the modeling and the control of an autonomous bicycle robot where the reference point is the center of gravity. The controls are based on the wheel heading's angular velocity and the steering's angular velocity. They have been developed to drive the autonomous bicycle robot from a given initial state to a final state, so that the total running cost is minimized. To solve the problem, the following approach was used: after having computed the control system Hamiltonian, Pontryagin's Minimum Principle was applied to derive the feasible controls and the costate system of ordinary differential equations. The feasible controls, derived as functions of the state and costate variables, were substituted into the combined nonlinear state–costate system of ordinary differential equations and yielded a control-free, state–costate system of ordinary differential equations. Such a system was judiciously vectorized to easily enable the application of any computer program written in Matlab, Octave or Scilab. A Matlab computer program, set as the main program, was developed to call a Runge–Kutta function coded into Matlab to solve the combined control-free, state–costate system of ordinary differential equations coded into a Matlab function. After running the program, the following results were obtained: seven feasible state functions from which the feasible trajectory of the robot is derived, seven feasible costate functions, and two feasible control functions. Computational simulations were developed and provided in order to persuade the readers of the effectiveness and the reliability of the approach.

**Keywords:** autonomous vehicle; bicycle robot; center of gravity; modeling; optimal control; path planning; differential equation; initial value problem; Runge–Kutta; scientific computing

## 1. Introduction

An autonomous bicycle is a high-efficiency vehicle robot that needs optimal management. It is easy and cheap to utilize, but its utilization involves significant and considerable challenges, which need to be addressed in very judicious, efficient and optimal ways. Some of the associated challenges concern its stability, its controllability, its observability, its robustness, etc. To overcome those challenges, robotics and automation have emerged as critical, current, relevant and innovative methods. Autonomous vehicles are becoming increasingly involved in humans' daily lives and create business opportunities for industries and research opportunities for universities. Researchers are undertaking modeling and computer simulation and many other experiments. The modeling and control of autonomous vehicles have become long-standing goals in applied mathematics, computer sciences and engineering. In each of the above-cited fields, researchers design research projects to address societal challenges and to meet societal demands. Research projects are also based on available funds as well as researchers' abilities, capabilities, experiences and expertise. There have been a considerable number of innovations and breakthroughs in robotics and automation, requiring governments to train people to not only behave as followers (in order to handle innovation) but also to behave as leaders regarding some aspects of innovation. In each of the research areas, work can be and is being carried out, and each study has its own hypothesis and objectives. Autonomous vehicle technology

impacts human life in a way that reduces transport costs, the number of car accidents and the number of deaths, as well as minimizing financial risk. Reports made by the transport departments of certain countries have shown that autonomous vehicles can minimize the number of deaths caused by car crashes [1], thus contributing to optimal fuel management and storage, contributing to the minimization of transport costs and saving approximately hundreds of billions of dollars for society as a whole. Despite the fact that robotics significantly and considerably impact human activities, human assistance during robot motion still remains relevant because of certain technical issues, which may suddenly arise during vehicle motion and are beyond the provided and embedded software and hardware.

There is work that has been carried out on the modeling and control of autonomous bicycles. Reference [2] aimed at designing an algorithm to self-stabilize an unmanned electric bicycle. Study [3] aimed at examining the impact of the reaction wheels on the stability of an autonomous bicycle. Reference [4] aimed at developing an under-actuated dynamic model for a tractor–trailer bicycle by using the Chaplygin equation. Inverse dynamics and a virtual prototype simulation were derived in order to prove the reliability of the developed dynamical model. Study [5] aimed at constructing a non-smooth dynamic model to plan the trajectory of a bicycle. A numerical solution algorithm was developed to show that a correct braking balance may considerably and significantly contribute to the stabilization of the vehicle. Reference [6] aimed at designing a path-following and a balance control algorithm for an autonomous motorcycle. Reference [7] aimed at studying the implications of braking style on the stability of a motorcycle. Stability analysis was performed and demonstrated that effective braking balance was able to considerably and significantly contribute to the stabilization of the vehicle. Study [8] aimed at designing a closed-loop control system for the motion of a bicycle. Reference [9] aimed at analyzing the dynamic model of autonomous bicycles using a control moment gyroscope (CMG). Reference [10] aimed at designing a steering controller to control bicycles under certain velocity conditions. Study [11] aimed at modelling an autonomous motorcycle and designing a control to follow a prescribed trajectory. Reference [12] aimed at modeling and designing control policies to stabilize a two-wheel bicycle. The experimental results confirmed that the bicycle robot was able to drive correctly. Reference [13] aimed at controlling an autonomous bicycle robot in which the slip is considered and an optimal feedback control is designed to drive the bicycle straight and stably. Reference [14] aimed at designing control policies to stabilize a two-wheeled autonomous vehicle. Study [15] aimed at developing a stationary self-sustaining two-wheeled vehicle. Reference [16] aimed at ameliorating an autonomous bicycle running stability in low-speed range. Study [17] aimed at examining the robust stabilization and path-following problems of riderless bicycles. Reference [18] aimed at developing a linear dynamic model for a bicycle robot with the purpose of driving the bicycle at a high speed. The open-loop stability was analyzed, and the controllability of the linear dynamic model was verified. Reference [19] developed an optimal control model to optimize the finacing operation. The associated dynamical system of the problem involves switching times. All the above-cited papers are relevant to the topic of this paper because they are explicitly or implicitly concerned with control.

To the best of my knowledge, no paper considers the control of an autonomous bicycle robot in which the optimal control theory methods are applied explicitly, and in which the state functions of the system are optimally predicted. When optimal control theory is applied, the system is weakened so that it can be handled easily. Most of time, assumptions are made to simplify and linearize a nonlinear system so that it can easily undergo certain operations, such as developing a quadratic regulator. However, when a control system is linearized and simplified, if no strategy is adopted to minimize the loss of information, the results will not be correct. In this paper, the system is highly nonlinear and has been examined as it is. No other paper has considered minimizing the running energy. No paper has considered the derivation of a new system consisting of the costate system of equations adjoining the state system. No paper has considered the impact of the adjoining system on the output of the state system. So, most of the

above-mentioned papers and those remaining in the list of references addressed the bicycle stability problem, the tracking problem, etc., without considering the minimization of the running costs. This paper focuses on the control of the kinematics of a bicycle robot where the reference point is the center of gravity. Such an autonomous bicycle is a highly nonlinear control system. The control of an autonomous bicycle is a rich topic in the way that it offers a number of significant, attractive and provocative problems associated with current research interests in the fields of mechatronics, applied mathematics, computer science, etc. An autonomous bicycle is a dynamical system subject to nonholonomic constraints on the contact wheels. It is naturally unstable. Its structure inspires a significant number of problems that researchers can address mathematically and computationally. In the literature, I discovered a set of rich and beautiful mathematical models requiring the reader to deal with a lot of mathematics. After combining these with my models, the kinematics is given by a nonlinear control system of seven ordinary differential equations involving seven state variables and two control variables. The state variables are the robot position with respect to the *x*- and the *y*-axis coordinates, the heading angle (which is the angle between the robot and the *x* axis), the steering angle, the heading angular velocity and the slipping angular velocity. The controls are based on the heading's angular acceleration and the steering's angular acceleration. Pontryagin's Minimum Principle is applied, and it derived the first-order necessary conditions for optimality from which a computer program is written and provided the feasible state functions, the feasible costate functions and the feasible control strategies.

The main contributions of this paper are the development of a mathematical model defined by the system of ordinary differential equations governing the reference commands (controls), the computation of the two feasible control strategies and the computation of the solution of the system defined by Equations (38)–(51). Such a solution gives seven feasible state functions (representing the feasible system responses to input controls) and seven costate (adjoint) functions. All the computations are performed by the three underlying computer programs developed in Matlab. The first computer program is used to code the system combining the state and the costate ordinary differential equations into a Matlab function. The second computer program is used to code a fourth-order Runge–Kutta numerical method into a Matlab function, and the third computer program, set as the main program, is used to code all the input and output operations as well as the operation consisting of calling the numerical method to solve the system of ordinary differential equations.

This paper is organized as follows: Section 2 develops different mathematical models and formulates the problem as an optimal control problem. Section 3 computes the Hamiltonian of the control system and solves the normal equations of optimality to obtain the expressions of the control functions. Section 4 applies Pontryagin's Minimum Principle to determine all relevant equations yielding the solutions. Section 5 develops relevant computer programs to determine the feasible control strategies, the corresponding feasible state functions and the feasible costate functions.

## 2. Mathematical Models

### 2.1. Objective Functional

In this paper, the total running energy to minimize is as follows:

$$J(\Omega) = J(\Omega_1, \Omega_2) = \int_{t_0}^{t_f} \left( \Omega_1{}^2 + \Omega_2{}^2 \right) dt \tag{1}$$

where $t_0$ and $t_f$ are the bicycle motion's start and end times, respectively. $\Omega_1 \in \mathbb{R}$ and $\Omega_2 \in \mathbb{R}$ are the reference commands regulating the bicycle heading angular velocity and the steering angular velocity, respectively, and correspond to the control variables. $h(t) = \Omega_1{}^2 + \Omega_2{}^2 \in \mathbb{R}^+$ is the cost rate.

### 2.2. Control System, Kinematic Model

The bicycle model is given by the following diagram:

Based on nonholonomic constraints and the below Geometric model given by Figure 1, the motion of an autonomous bicycle is kinematically modelled as follows:

$$\frac{dx}{dt} = c_1 \omega cos(\theta + \beta) \tag{2}$$

$$\frac{dy}{dt} = c_1 \omega sin(\theta + \beta) \tag{3}$$

$$\frac{d\theta}{dt} = c_3 \omega tan(\delta)cos(\beta) \tag{4}$$

$$\frac{d\delta}{dt} = \varphi \tag{5}$$

$$\frac{d\beta}{dt} = c_2 \varphi (cos(\beta) / cos(\delta))^2 \tag{6}$$

where $c_1 = R \in \mathbb{R}^+$, $c_2 = \frac{l_r}{L} \in \mathbb{R}^+$ and $c_3 = \frac{R}{L} \in \mathbb{R}^+$ are the constants of proportionality where $R$, $L$ and $l_r$ with ( $R < l_r < L$) are the radius of each wheel, the distance between the centre of the rear wheel and the center of the front wheel and the distance between the rear-axle and the center of gravity, respectively. $(x, y)$ are the coordinates of the projection of the center of gravity on the horizontal plane, $\theta \in \mathbb{R}$ is the heading angle, $\delta \in \mathbb{R}$ is the steering angle, $\beta \in \mathbb{R}$ is the slip angle, $\varphi \in \mathbb{R}$ is the steering angular velocity and $\omega \in \mathbb{R}$ is the heading angular velocity of the wheels, and $S$ is the length of the left-hand side of the bigger triangle.

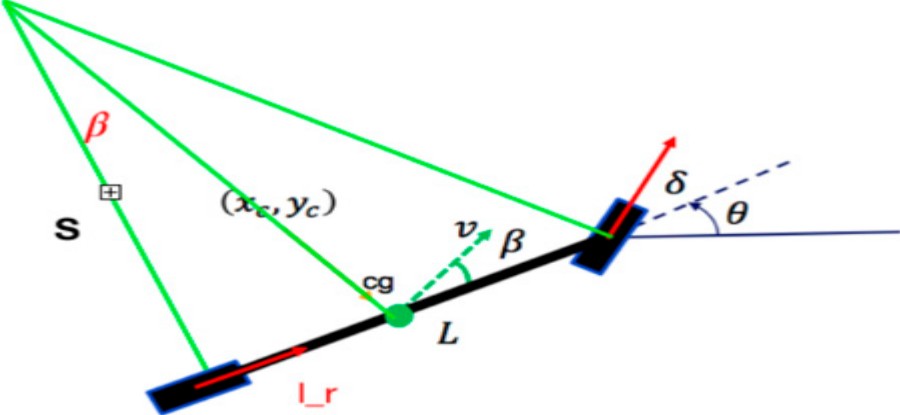

**Figure 1.** Geometric Model of a Center of Gravity-based autonomous Bicycle Robot.

The reference commands which regulate the bicycle's wheels' angular velocity and the slip angular velocity are given by the following closed-loop system of ordinary differential equations:

$$\frac{d\omega}{dt} = -a_1 \omega + a_1 \Omega_1 \tag{7}$$

$$\frac{d\varphi}{dt} = -a_2 \varphi + a_2 \Omega_2 \tag{8}$$

where $\Omega_1$ and $\Omega_2$ are the reference commands, and $a_1$ and $a_2$ are parameters of proportionality.

By letting $z_1 = x$, $z_2 = y$, $z_3 = \theta$, $z_4 = \delta$, $z_5 = \beta$, $z_6 = \omega$ and $z_7 = \varphi$ be state variables and $\Omega_1$ and $\Omega_2$ be control variables, then the above models can be reformulated as:

$$J(\Omega) = J(\Omega_1, \Omega_2) = \int_{t_0}^{t_f} \left( \Omega_1{}^2 + \Omega_2{}^2 \right) dt \tag{9}$$

and

$$\frac{dz_1}{dt} = c_1 z_6 cos(z_3 + z_5) \tag{10}$$

$$\frac{dz_2}{dt} = c_1 z_6 sin(z_3 + z_5) \tag{11}$$

$$\frac{dz_3}{dt} = c_3 z_6 tan(z_4) cos(z_5) \tag{12}$$

$$\frac{dz_4}{dt} = z_7 \tag{13}$$

$$\frac{dz_5}{dt} = c_2 z_7 cos^2(z_5) / cos^2(z_4) \tag{14}$$

$$\frac{dz_6}{dt} = -a_1 z_6 + a_1 \Omega_1 \tag{15}$$

$$\frac{dz_7}{dt} = -a_2 z_7 + a_2 \Omega_2 \tag{16}$$

*2.3. Problem Formulation*

From all that precedes, the problem can be formulated as follows:

For a given time interval $\left[t_0, t_f\right] \subset \mathbb{R}^+$ and initial states $z_1(0) = z_{01}$, $z_2(0) = z_{02}$, $z_3(0) = z_{03}$, $z_4(0) = z_{04}$, $z_5(0) = z_{05}$, $z_6(0) = z_{06}$ and $z_7(0) = z_{07}$, compute the feasible control strategies $\Omega_1(t)$ and $\Omega_2(t)$ as well as the corresponding feasible system responses, which must be defined by the state functions $z_1(t)$, $z_2(t)$, $z_3(t)$, $z_4(t)$, $z_5(t)$, $z_6(t)$ and $z_7(t)$, so that the autonomous bicycle robot can move from a given state to a final state such that the total running cost of energy spent by the bicycle is minimized. The kinematics of the robot's motion is then given by the system of ordinary differential Equations (10)–(16). Optimal control theory emerges as a relevant approach to solve the problem (9)–(16) because it has many strong and convincing theorems. We will focus on the design of open-loop control strategies.

## 3. Hamiltonian and Feasible Controls

The Hamiltonian of the system is given by:

$$H(t, \mathbf{Y}(t), \boldsymbol{\alpha}(t), \Omega_1(t), \Omega_2(t)) = h(t) + \sum_{k=1}^{7} \alpha_k(t) f_k(\mathbf{Y}(t), \Omega_1(t), \Omega_2(t)) \tag{17}$$

where:

$h(t) = \Omega_1{}^2 + \Omega_2{}^2$ is energy cost rate;

$f_1(\mathbf{Y}(t), \Omega_1(t), \Omega_2(t)) = c_1 z_6 cos(z_3 + z_5)$ is the $x$ component of the linear velocity of the bicycle;

$f_2(\mathbf{Y}(t), \Omega_1(t), \Omega_2(t)) = c_1 z_6 sin(z_3 + z_5)$ is the $y$ component of the linear velocity of the bicycle;

$f_3(\mathbf{Y}(t), \Omega_1(t), \Omega_2(t)) = c_3 z_6 tan(z_4) cos(z_5)$ is the heading angular velocity;

$f_4(\mathbf{Y}(t), \Omega_1(t), \Omega_2(t)) = z_7$ is the steering angular velocity;

$f_5(\mathbf{Y}(t), \Omega_1(t), \Omega_2(t)) = c_2 z_7 cos^2(z_5) / cos^2(z_4)$ is the slipping angular velocity;

$f_6(\mathbf{Y}(t), \Omega_1(t), \Omega_2(t)) = -a_1 z_6 + a_1 \Omega_1$ is the bicycle heading acceleration;

$f_7(\mathbf{Y}(t), \Omega_1(t), \Omega_2(t)) = -a_2 z_7 + a_2 \Omega_2$ is the bicycle steering acceleration;

$\mathbf{Y}(t) = (x(t), y(t), \theta(t), \delta(t), \beta(t), \omega(t), \varphi(t)) = (z_1(t), z_2(t), z_3(t), z_4(t), z_5(t), z_6(t), z_7(t))$ is the unknown state vector function;

$\boldsymbol{\alpha}(t) = (\alpha_1(t), \alpha_2(t), \alpha_3(t), \alpha_4(t), \alpha_5(t), \alpha_6(t), \alpha_7(t))$ is the unkown costate (adjoint) vector function; $\Omega = (\Omega_1, \Omega_2)$ is the control vector.

Each of the functions $f_i$ are continuous on $\left[t_0, t_f\right]$.

Define $z_8 = \alpha_1$, $z_9 = \alpha_2$, $z_{10} = \alpha_3$, $z_{11} = \alpha_4$, $z_{12} = \alpha_5$, $z_{13} = \alpha_6$ and $z_{14} = \alpha_7$.

The feasible controls' normal equations for optimality are as follows:

$$\frac{\partial H}{\partial \Omega_1} = 2\Omega_1{}^* + a_1\alpha_6 = 0 \tag{18}$$

and

$$\frac{\partial H}{\partial \Omega_2} = 2\Omega_2{}^* + a_2\alpha_7 = 0 \tag{19}$$

The feasible controls are given by:

$$\Omega_1{}^* = -0.5a_1\alpha_6 \tag{20}$$

and

$$\Omega_2{}^* = -0.5a_2\alpha_7 \tag{21}$$

In terms of new variables, the controls' normal equations are:

$$\frac{\partial H}{\partial \Omega_1} = 2\Omega_1{}^* + a_1 z_{13} = 0 \tag{22}$$

and

$$\frac{\partial H}{\partial \Omega_2} = 2\Omega_2{}^* + a_2 z_{14} = 0 \tag{23}$$

The feasible controls are given by:

$$\Omega_1{}^* = -0.5a_1 z_{13} \tag{24}$$

and

$$\Omega_2{}^* = -0.5a_2 z_{14} \tag{25}$$

From (24), notice that the first control strategy function $\Omega_1{}^* = -0.5a_1 z_{13}$ involves the costate variable $z_{13}$; $z_{13}$ is the costate variable adjoint and connected to variable $z_6$. The variable $z_6$ is involved in the motion of the robot along the $x$ and $y$ axis. $x$ and $y$ constitute the driving parameters of the autonomous bicycle. Also notice from (25) that the second control function $\Omega_2{}^* = -0.5a_1 z_{14}$ involves the costate variable $z_{14}$; $z_{14}$ is the costate variable adjoint and connected to variable $z_7$, which is connected to $z_4$.

## 4. Pontryagin's Minimum Principle

Pontryagin's Minimum Principle is a method based on the calculus of variations to solve optimal control problems with state constraints.

Optimal Control Theory is applied in finance [18,20], engineering [21], biology, geology, image processing, etc. Some of the problems solved through Pontryagin's Minimum Principle (PMP) are the Minimum Time Problem, Minimum Fuel Problem, Minimum Energy Problem, Minimum Risk Problem, Minimum Cost of Operation and Minimum Loss Problem. The developed results are for optimal, effective and reliable decision making.

*Theorem*

The necessary conditions for the pair $(x^*(t), u^*(t))$ to be optimal [11] is the existence of a costate vector $\alpha^*(t)$ such that:

$$\frac{dx^*}{dt}(t) = \frac{\partial H}{\partial \alpha}(t, x^*(t), u^*(t), \alpha^*(t)) \tag{26}$$

$$\frac{d\alpha^*}{dt}(t) = -\frac{\partial H}{\partial \alpha}(t, x^*(t), u^*(t), \alpha^*(t)) \tag{27}$$

$$H(t, x^*(t), u^*(t), \alpha^*(t)) \le H(t, x^*(t), u(t), \alpha^*(t)) \tag{28}$$

and the following boundary condition:

$$\left[\frac{\partial h}{\partial x}\left(t_f, x^*\left(t_f\right)\right) - \alpha^*\left(t_f\right)\right]^T \delta x_f + \left[H\left(t, x^*\left(t_f\right), u^*\left(t_f\right), \alpha^*\left(t_f\right)\right) + \frac{\partial h}{\partial x}\left(t_f, x^*\left(t_f\right)\right)\right]\delta t_f = 0 \tag{29}$$

$u^*(t)$ is a control that minimizes $H(t, x^*(t), u(t), t, \alpha^*(t))$. Notice that the above conditions are necessary and not sufficient. Thus, we can make a conclusion that a necessary condition for a control to minimize the performance cost function J is the following

$H(t, x^*(t), u^*(t), \alpha^*(t)) \leq H(t, x^*(t), u(t), \alpha^*(t))$ for all $t \in \left[t_0, t_f\right]$ and for all admissible controls $u$.

The above theorem is called Pontryagin's Minimum Principle.

From all that precedes, the application of Pontryagin's Minimum Principle to this article's problem yields the following:

If $\Omega^* = (\Omega_1{}^*, \Omega_2{}^*) \in \mathbb{R}^2$ is the feasible control vector of the above problem and $Y^* = (z_1{}^*, z_2{}^*, z_3{}^*, z_4{}^*, z_5{}^*, z_6{}^*, z_7{}^*) \in \mathbb{R}^7$ is the associated feasible system response, then there exists a costate vector $\alpha^* = (z_8{}^*, z_9{}^*, z_{10}{}^*, z_{11}{}^*, z_{12}{}^*, z_{13}{}^*, z_{14}{}^*) \in \mathbb{R}^7$ such that the following properties are satisfied:

$$J(\Omega^*) \leq J(\Omega) \forall \Omega \in \mathbb{R}^2 \tag{30}$$

$$\frac{dz_8}{dt} = -\frac{\partial H}{\partial z_1} = 0 \tag{31}$$

$$\frac{dz_9}{dt} = -\frac{\partial H}{\partial z_2} = 0 \tag{32}$$

$$\frac{dz_{10}}{dt} = -\frac{\partial H}{\partial z_3} = c_1 z_6 (z_8 \sin(z_3 + z_5) - z_9 \cos(z_3 + z_5)) \tag{33}$$

$$\frac{dz_{11}}{dt} = -\frac{\partial H}{\partial z_4} = -c_3 z_6 z_{10} \cos(z_5) \sec^2(z_4) + 2c_2 z_7 z_{12} \cos^2(z_5) \tan(z_4) \sec^2(z_4) \tag{34}$$

$$\frac{dz_{12}}{dt} = -\frac{\partial H}{\partial z_5} = c_1 z_6 (z_8 \sin(z_3 + z_5) - z_9 \cos(z_3 + z_5)) + c_3 z_6 z_{10} \tan(z_4) \sin(z_5) + l_2 z_7 z_{12} \cos^{-2}(z_4) \sin(2z_6) \tag{35}$$

$$\frac{dz_{13}}{dt} = -\frac{\partial H}{\partial z_6} = -c_1(z_8 \cos(z_3 + z_5) + z_9 \sin(z_3 + z_5)) - c_3 z_{10} \tan(z_4) \cos(z_5) + a_1 z_{13} \tag{36}$$

$$\frac{dz_{14}}{dt} = -\frac{\partial H}{\partial z_7} = -z_{11} - c_2 z_{12} \cos^2(z_5)/\cos^2(z_4) + a_2 z_{14} \tag{37}$$

Equation (30) is the optimality condition for the control strategies, and the system of Equations (31)–(37) represent the costate system. By combining the state and the costate systems into a vector as $z = [Y, \alpha]$, and by substituting $\Omega_1{}^* = -0.5a_1 z_{13}$ and $\Omega_2{}^* = -0.5a_2 z_{14}$ into the system, which is being built, then the combined state–costate system can be rewritten as follows:

$$\frac{dz_1}{dt} = c_1 z_6 cos(z_3 + z_5) \tag{38}$$

$$\frac{dz_2}{dt} = c_1 z_6 sin(z_3 + z_5) \tag{39}$$

$$\frac{dz_3}{dt} = c_3 z_6 tan(z_4) cos(z_5) \tag{40}$$

$$\frac{dz_4}{dt} = z_7 \tag{41}$$

$$\frac{dz_5}{dt} = c_2 z_7 cos^2(z_5)/cos^2(z_4) \tag{42}$$

$$\frac{dz_6}{dt} = -a_1 z_6 + a_1(-0.5a_1 z_{13}) \tag{43}$$

$$\frac{dz_7}{dt} = -a_2 z_7 + a_2(-0.5 a_2 z_{14}) \tag{44}$$

$$\frac{dz_8}{dt} = 0 \tag{45}$$

$$\frac{dz_9}{dt} = 0 \tag{46}$$

$$\frac{dz_{10}}{dt} = c_1 z_6 (z_8 \sin(z_3 + z_5) - z_9 \cos(z_3 + z_5)) \tag{47}$$

$$\frac{dz_{11}}{dt} = -c_3 z_6 z_{10} \cos(z_5) \sec^2(z_4) + 2c_2 z_7 z_{12} \cos^2(z_5) \tan(z_4) \sec^2(z_4) \tag{48}$$

$$\frac{dz_{12}}{dt} = c_1 z_6 (z_8 \sin(z_3 + z_5) - z_9 \cos(z_3 + z_5)) + c_3 z_6 z_{10} \tan(z_4) \sin(z_5) + c_2 z_7 z_{12} \cos^{-2}(z_4) \sin(2z_6) \tag{49}$$

$$\frac{dz_{13}}{dt} = -c_1 (z_8 \cos(z_3 + z_5) + z_9 \sin(z_3 + z_5)) - c_3 z_{10} \tan(z_4) \cos(z_5) + a_1 z_{13} \tag{50}$$

$$\frac{dz_{14}}{dt} = -z_{11} - c_2 z_{12} cos^2(z_5)/cos^2(z_4) + a_2 z_{14} \tag{51}$$

## 5. Numerical and Computational Simulations

The above system (38)–(51) can be vectorized as $\frac{d\mathbf{Z}}{dt} = f(t, \mathbf{Z})$ (52), where $\mathbf{Z} = [z_1, z_2, z_3, z_4, z_5, z_6, z_7, z_8, z_9, z_{10}, z_{11}, z_{12}, z_{13}, z_{14}]^T \in \mathbb{R}^{14}$ is the vector storing seven state variables and seven costate variables. The following parameters are used:

$$R = 0.4; \ L = 0.8; \ l_r = 0.4; \ c_1 = R; \ c_2 = \frac{l_r}{L}; \ c_3 = \frac{R}{L}; \ a_1 = 0.25; \ a_2 = 0.25; \ t_0 = 0;$$

$$t_f = 5; \ N = 501$$

where $t_0$ is the lower bound of the time interval, $t_f$ is the upper bound of the time interval and $N$ is the number of discrete points on the interval $\left[t_0, t_f\right]$. The remaining parameters are defined in the previous sections.

In order to develop an algorithm which can solve any system of ordinary differential equations, we wrote a Matlab function that is an algorithm coding a fourth-order Runge–Kutta method. Such an Algorithms 1–4 is as follows:

---
**Algorithm 1** Fourth-order Runge–Kutta method

---
```
function [t,z] = runge_v2(state_costate_robot,t0,tf,N,z0)
h = (tf–t0)./(N−1); % N is the number of discrete points, h is the step.
% t0 and tf are the lower bound and the upper bound of the time interval [t0,tf].
t = t0:h:tf; % Discretization of the time interval [t0,tf].
% t is the time vector with N elements. His elements are the discrete time points. t = t′;
z = zeros(N,length(z0)); % z is initialized to zero. It is initially set as a matrix of N rows
and with the
The first row of the solution z is set to z0;
for n = 2:N
k1 = feval(fs,t(n − 1),z(n − 1,:));
k2 = feval(fs,t(n − 1) + (h/2),z(n − 1,:) + (h/2) × k1′);
k3 = feval(fs,t(n − 1) + (h/2),z(n − 1,:) + (h/2) × k2′);
k4 = feval(fs,t(n − 1) + h,z(n − 1,:) + h × k3′);
z(n,:) = z(n − 1,:) + (h/6) × (k1′ + 2 × k2′ + 2 × k3′+k4′);
end
```
---

The above algorithm can be used to solve any initial value problem. Let us use it to solve the above combined state–costate system of ordinary differential Equations (38)–(51).

The combined state–costate system of ordinary differential equations is coded into the following vector function:

---

**Algorithm 2** The combined state–costate system of ordinary differential equations

---

```
function dzdt = bicycle_centerOfGravity (t,z)
dzdt = zeros(14,1);
R = 0.4; L = 0.8; lr = 0.4; c1 = R; c2 = lr/L; c3 = R/L; a1 = 0.25; a2 = 0.25;
dzdt (1) = c1 × z(6) × cos(z(3) + z(5)); %Coding of Equation (38)
dzdt (2) = c1 × z(6) × sin(z(3) + z(5)); % Coding of Equation (39)
dzdt (3) = c3 × z(6) × tan(z(4)) × cos((z(5))); % Coding of Equation (40)
dzdt (4) = z(7); % Coding of Equation (41)
dzdt (5) = c2 × z(7) × ((cos(z(5))^2)/(cos(z(4))^2)); % Coding of Equation (42)
dzdt (6) = −a1 × z(6) + a1 × (−0.5 × a1 × z(13)); % Coding of Equation (43)
dzdt (7) = −a2 × z(7) + a2 × (−0.5 × a2 × z(14)); % Coding of Equation (44)
dzdt (8) = 0; % Coding of Equation (45)
dzdt (9) = 0; % Coding of Equation (46)
dzdt (10) = c1 × z(6) × (z(8) × sin(z(3) + z(5)) − z(9) × cos(z(3) + z(5))); % Coding of Equation (47)
dzdt (11) = −c3 × z(6) × z(10) × cos(z(5)) × (sec(z(4))^2) + 2 × c2 × z(7) × z(12) × (cos(z(5))^2) ×
tan(z(4)) × (sec(z(4))^2); % Coding of Equation (48)
dzdt (12) = c1 × z(6) × (z(8) × sin(z(3) + z(5)) − z(9) × cos(z(3) + z(5))) + c3 × z(6) × z(10) ×
tan(z(4)) × sin(z(5)) + c2 × z(7) × z(12) × (cos(z(4))^(−2)) × sin(2 × z(6)); % Coding of
Equation (49)
dzdt (13) = −c1 × (z(8) × cos(z(3) + z(5)) + z(9) × sin(z(3) + z(5))) −c3 × z(10) × tan(z(4)) ×
cos(z(5)) + a1 × z(13);
% Coding of Equation (50)
dzdt (14) = −z(11) − c2 × z(12) × ((cos(z(5))/cos(z(4)))^2) + a2 × z(14); % Coding of Equation (51)
```

---

The function coding the state–costate system of ordinary differential equations can also be written as:

---

**Algorithm 3** State–costate system of ordinary differential equations

---

```
function dzdt = bicycle_centerOfGravity (t,z)
dzdt = zeros(14,1); R = 0.4; L = 0.8; lr = 0.4; c1 = R; c2 = lr/L; c3 = R/L; a1 = 0.25; a2 = 0.25;
dzdt = [c1 × z(6) × cos(z(3) + z(5));
c1 × z(6) × sin(z(3) + z(5));
c3 × z(6) × tan(z(4)) × cos((z(5)));
z(7);
c2 × z(7) × ((cos(z(5))^2)/(cos(z(4))^2));
−a1 × z(6) + a1 × (−0.5 × a1 × z(13));
−a2 × z(7) + a2 × (−0.5 × a2 × z(14));
0;
0;
c1 × z(6) × (z(8) × sin(z(3) + z(5)) − z(9) × cos(z(3) + z(5)));
−c3 × z(6) × z(10) × cos(z(5)) × (sec(z(4))^2) + 2 × c2 × z(7) × z(12) × (cos(z(5))^2) × tan(z(4)) ×
(sec(z(4))^2);
c1 × z(6) × (z(8) × sin(z(3)+z(5)) − z(9) × cos(z(3)+z(5))) + c3 × z(6) × z(10) × tan(z(4)) ×
sin(z(5))
+c2 × z(7) × z(12) × (cos(z(4))^(−2)) × sin(2 × z(6));
−c1 × (z(8) × cos(z(3) + z(5)) + z(9) × sin(z(3) + z(5))) − c3 × z(10) × tan(z(4)) × cos(z(5)) + a1 ×
z(13);
−z(11) −c2 × z(12) × ((cos(z(5))/cos(z(4)))^2) + a2 × z(14)];
```

---

Three different Matlab functions, "function [t,z] = runge_kutta(fs,t0,tf,N,z0) ", "function dzdt = bicycle_centerOfGravity (t,z) " and "function main_ bicycle_centerOfGravity", are developed in three different files. main_ bicycle_centerOfGravity is a script function. For sake of ease, let us name these tree files runge_kutta.m, bicycle_centerOfGravity.m and

main_ bicycle_centerOfGravity.m, respectively, where main_ bicycle_centerOfGravity.m is the main file.

As said above, function dzdt = bicycle_centerOfGravity (t,z) codes the system combining the state and the costate's ordinary differential equations, and [t,z] = runge_v2(fs,t0,tf,N,z0) codes a fourth-order Runge–Kutta numerical method to solve any system of ordinary differential equations. In addition to statements of allocation, the script file contains also a statement calling.

function [t,z] = runge_v2(fs,t0,tf,N,z0) to solve the system of differential equations defined by function.

dzdt = bicycle_centerOfGravity (t,z). The system is a first-order system. The main file is given by the following set of codes:

---

**Algorithm 4** First-order system

---

```
function main_bicycle_centerOfGravity
clear all
clc
format long
disp('Nonlinear Control of an Autonomous Bicycle Robot: Computation of Feasible Controls')
R = 0.4; L = 0.8; lr = 0.4; c1 = R; c2 = lr/L; c3 = R/L; a1 = 0.25; a2 = 0.25; t0 = 0; tf = 5; N = 501;
z0 = [zeros (7,1);2 * ones (7,1)];
[t,z] = runge_v2(' bicycle_centerOfGravity ',t0,tf,N,z0);
control1 = −0.5 × a1 × z(:,13); control2 = −0.5 × a2 × z(:,14); control = [ control1, control2];
dx = c1 × z(6) × cos(z(3) + z(5)); % x component of the velocity
dy = c1 × z(6) × sin(z(3) + z(5)); % y component of the velocity
dTheta = c3 × z(6) × tan(z(4)) × cos((z(5))); % Heading angular velocity
dDelta = z(7); % Steering angular velocity
dOmega = c2 × z(7) × ((cos(z(5))^2)/(cos(z(4))^2)); % Rate of change of the slipping angular
velocity
dPhi = −a1 × z(6) + a1 × (−0.5 × a1 × z(13));
dzdt7 = −a2 × z(7) + a2 × (−0.5 × a2 × z(14));
% Feasible trajectory
plot(z(:,1),z(:,2),'r'); xlabel('x (in meters)');ylabel('y = f(x) (in meters)');
% Converting the plot into a png (Portable Network Graphic) format
print C:\Users\Guest\Documents\16september2022\bicyclepath.png;
disp('To display the first three state functions, press a key')
% distance = intsplin(t,sqrt(dx.^2 + dy.^2));
% First 3 state functions
subplot(3,1,1); plot(t,z(:,1),'r'); xlabel('Time t in seconds'); ylabel('State 1 ');
subplot(3,1,2); plot(t,z(:,2),'r'); xlabel('Time t in seconds '); ylabel('State 2 ');
subplot(3,1,3); plot(t,z(:,3),'r'); xlabel('Time t in seconds '); ylabel('State 3 ');
% The file containing the first three state function graphs is named "bicycleFirst3states" and saved
in the folder whose path is C:\Users\Guest\Documents\16september2022
% Converting the plot into png (Portable Graphic network) format
print C:\Users\Guest\Documents\16september2022\bicycleFirst3states.png;
% Last 4 state functions
disp('To display the last four state functions, press a key')
subplot(2,2,1); plot(t,z(:,4),'r'); xlabel('Time t in seconds'); ylabel('State 4 ');
subplot(2,2,2); plot(t,z(:,5),'r'); xlabel('Time t in seconds '); ylabel('State 5 ');
subplot(2,2,3); plot(t,z(:,6),'r'); xlabel('Time t in seconds '); ylabel('State 6 ');
subplot(2,2,4); plot(t,z(:,7),'r'); xlabel('Time t in seconds '); ylabel('State 7 ');
% Converting the plot into png (Portable Graphic network) format
print C:\Users\Guest\Documents\16september2022\bicycleLast4states.png;
% First 3 costate functions
disp('To display the first three costate functions, press a key')
subplot(3,1,1); plot(t,z(:,8),'r'); xlabel('Time t in seconds'); ylabel('Costate 1 ');
subplot(3,1,2); plot(t,z(:,9),'r'); xlabel('Time t in seconds '); ylabel('Costate 2 '); subplot(3,1,3);
plot(t,z(:,10),'r'); xlabel('Time t in seconds '); ylabel('Costate 3 ');
```

---

---

**Algorithm 4** First-order system

---

```
% Converting the plot into png (Portable Graphic network) format
print C:\Users\Guest\Documents\16september2022\bicycleFirst3Costates.png;
% Last 4 costate functions
disp('To display the last four costate functions, press a key')
subplot(2,2,1); plot(t,z(:,11),'r'); xlabel('Time t in seconds'); ylabel('Costate 4');
subplot(2,2,2); plot(t,z(:,12),'r'); xlabel('Time t in seconds '); ylabel('Costate 5');
subplot(2,2,3); plot(t,z(:,13),'r'); xlabel('Time t in seconds '); ylabel('Costate 6');
subplot(2,2,4); plot(t,z(:,14),'r'); xlabel('Time t in seconds '); ylabel('Costate 7');
% Converting the plot into png (Portable Graphic network) format
print C:\Users\Guest\Documents\16september2022\bicycleLast4Costates.png;
% Control strategies
disp('To display the two control functions, press a key')
subplot(2,1,1); plot(t,control1,'r'); xlabel('Time t in seconds '); ylabel('Control1');
subplot(2,1,2); plot(t,control2,'r'); xlabel('Time t in seconds '); ylabel('Control2');
% Converting the plot into png (Portable Graphic network) format
print C:\Users\Guest\Documents\16september2022\bicycleControls.png
% Velocities
disp('To display the directional velocities functions, press a key')
subplot(3,1,1); plot(t,dx,'r'); xlabel('Time t in seconds '); ylabel('x velocity');
subplot(3,1,2); plot(t,dy,'r'); xlabel('Time t in seconds '); ylabel('y velocity ');
subplot(3,1,3); plot(t,c1*abs(z(:,6)),'r'); xlabel('Time t in seconds ');ylabel('Robot speed');
% Converting the plot into png (Portable Graphic network) format
print C:\Users\Guest\Documents\16september2022\bicycle_velocities.png
diary C:\Users\Guest\Documents\16september2022\Results_Runge-Kutta;
% diary allows to store the numerical data and results into a specified file.
```

---

**Case 1:** The initial heading angle for the position of the robot is 0 radian; the other parameters also are 0.

This means that the initial condition is z0 = [zeros (7,1);2 ∗ ones (7,1)], in which zeros (7,1) is the initial state vector and 2 ∗ ones (7,1) is the initial costate vector. The following are the results: Figure 2 presents the trajectory of the autonomous bicycle robot, where $(0,0)$ is the starting point, and 0 is the starting heading angle between the robot and the $x$ axis. Figure 2 is plotted in the third quadrant. For Figure 3, the magnitude of the initial costate vector is increased. Then, one can notice that the length of the resulting robot's trajectory has increased. One can also verify with the program that, in the initial condition, if the costate vector 25 ∗ ones (7,1) is replaced by the vector −25 ∗ ones (7,1), the curve will be displaced from the third quadrant to the fourth quadrant with the same length.

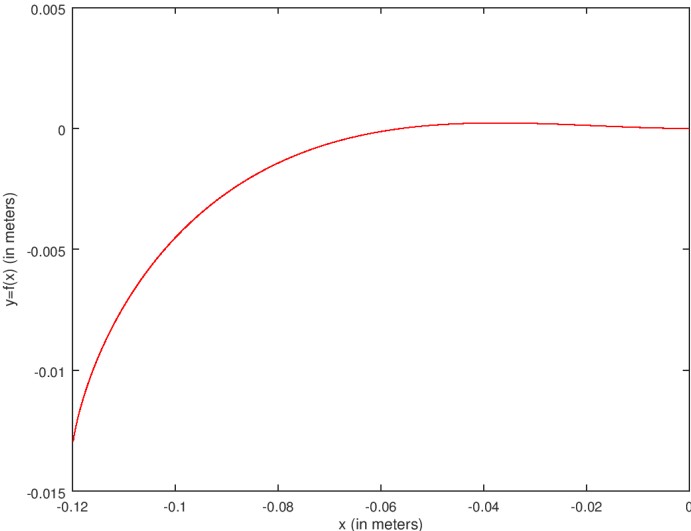

**Figure 2.** Feasible Vehicle Robot Trajectory (Initial condition = [zeros (7,1); 2 ∗ ones (7,1)]).

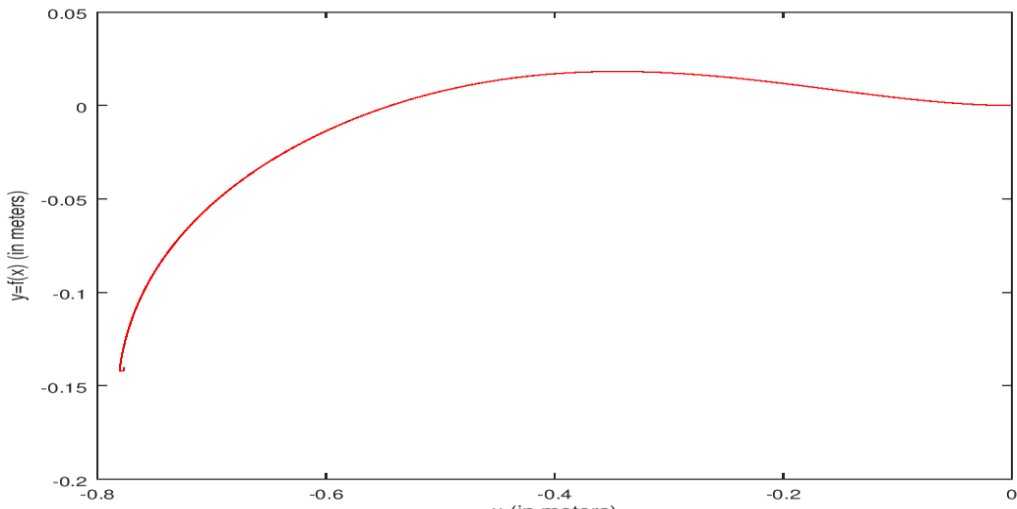

**Figure 3.** Feasible Vehicle Robot Trajectory (Initial condition = [zeros (7,1); 25 ∗ ones (7,1)]).

Figure 4 presents the first three state functions $x(t)$, $y(t)$ and $\theta(t)$ defined, respectively, by state 1, state 2 and state 3. The points $(x(t), y(t))$ defines the above feasible trajectory of the robot. In this case, for the starting position of the robot $(x(t = 0), y(t = 0)) = (0,0)$, $\theta(t = 0) = 0$, We have the following:

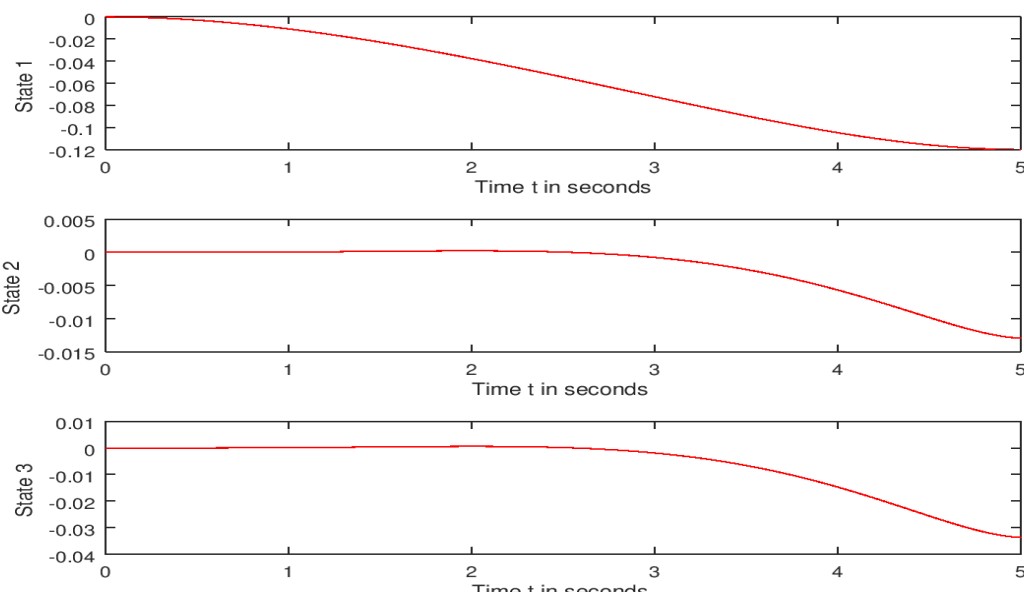

**Figure 4.** Feasible first three state functions.

Figure 5 presents the last four state functions $\delta(t)$, $\beta(t)$, $\omega(t)$ and $\varphi(t)$ given in the graph by State 4 (the steering angle), State 5 (the slip angle), State 6 (the steering angular velocity) and State 7 (the heading angular velocity), respectively. Notice that state 4, state 5 and state 7 are increasing.

Figure 6 presents the first three costate functions $\alpha_1(t)$, $\alpha_2(t)$ and $\alpha_3(t)$ (adjoint functions to $x(t)$, $y(t)$ and $\theta(t)$, respectively) given, respectively, by Costate 1, Costate 2 and Costate 3. One can notice that the first two costate functions are constant functions due to the values of their time derivatives, which are zero, given by the Equations (41) and (42).

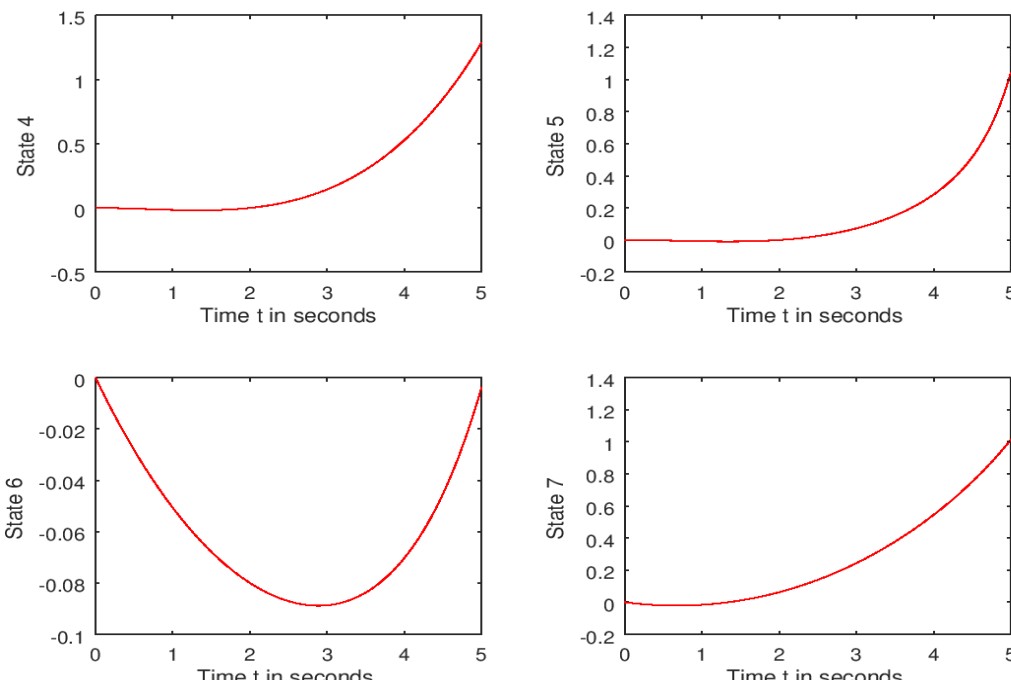

**Figure 5.** Feasible last four state functions.

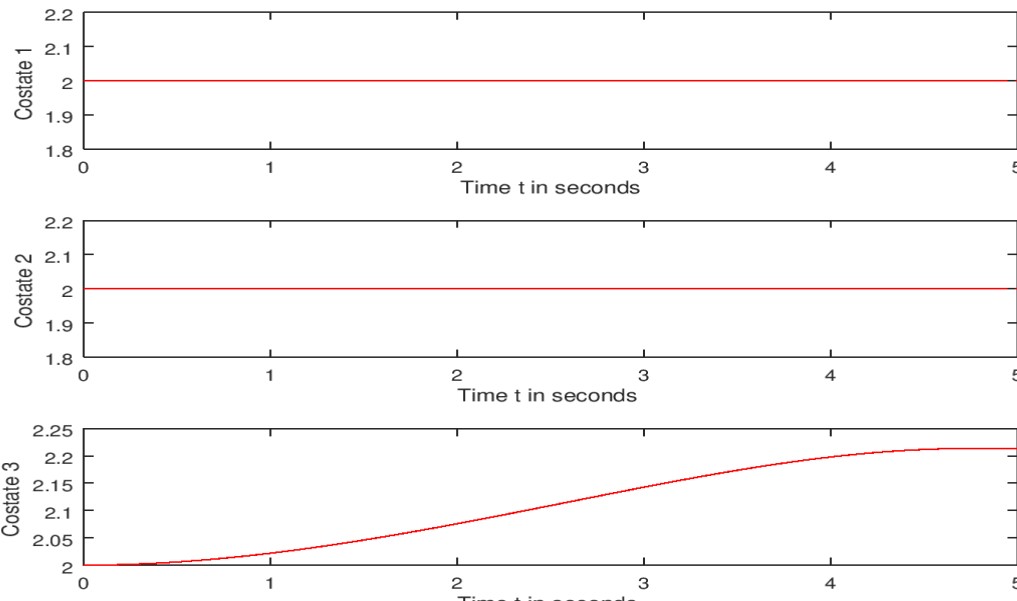

**Figure 6.** Feasible first three costate functions.

Figure 7 presents the last four costate functions $\alpha_4(t)$, $\alpha_5(t)$, $\alpha_6(t)$ and $\alpha_7(t)$ (adjoint functions to $\delta(t)$, $\beta(t)$, $\omega(t)$ and $\varphi(t)$, respectively) given, respectively, by Costate 4, Costate 5, Costate 6 and Costate 7.

Figure 8 presents the feasible controls. One can notice that the two control functions are increasing in the given interval. Control 1 and Control 2 are based on the bicycle wheels' heading angular velocity and slip angular velocity, respectively.

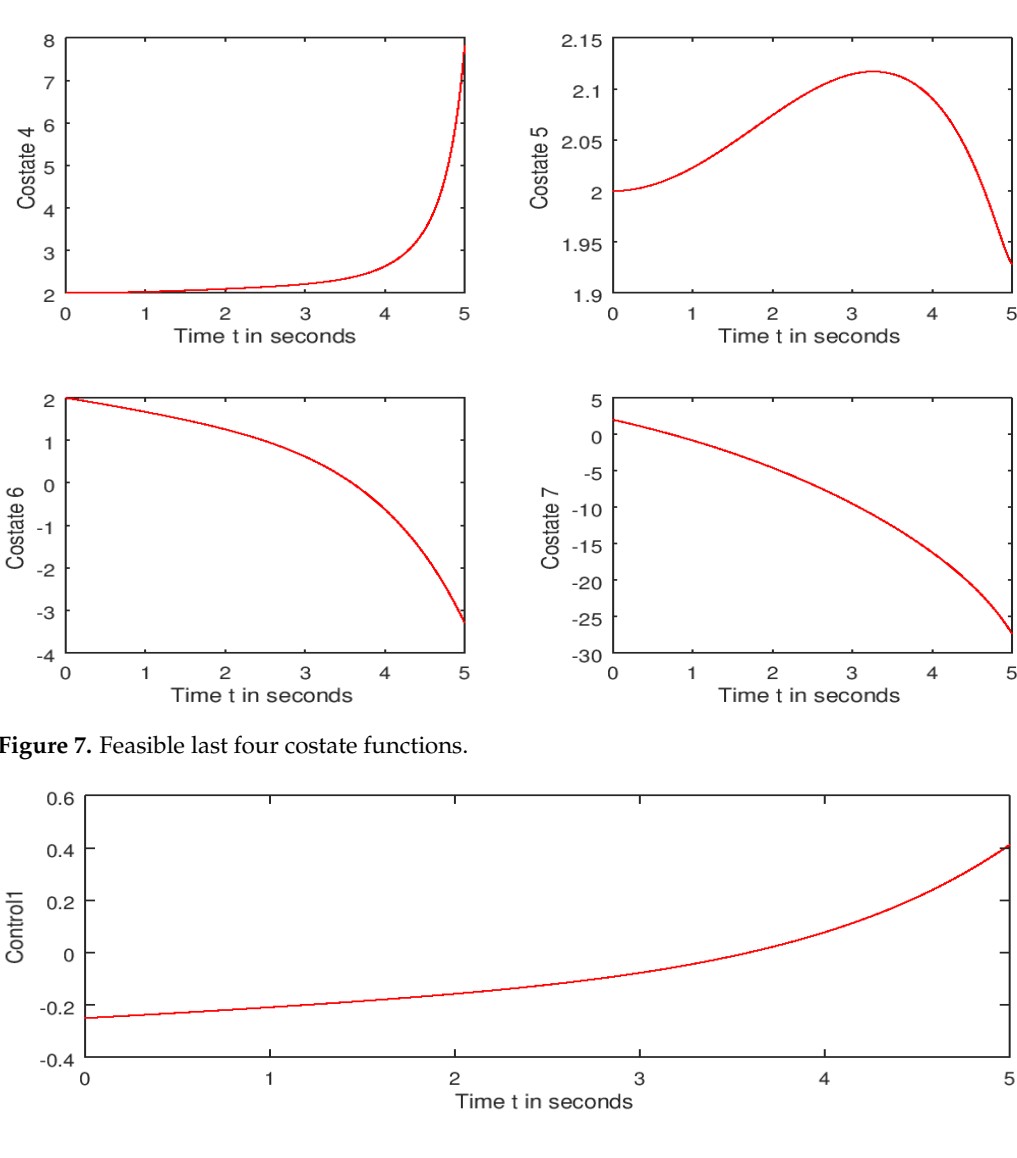

**Figure 7.** Feasible last four costate functions.

**Figure 8.** Feasible Control functions.

Figure 9 presents the velocity along the $x$ direction, the velocity along the $y$ direction and the feasible speed. One can notice that the speed increases in a time interval and decreases in another time interval. Notice from the above figure that each control strategy is increasing.

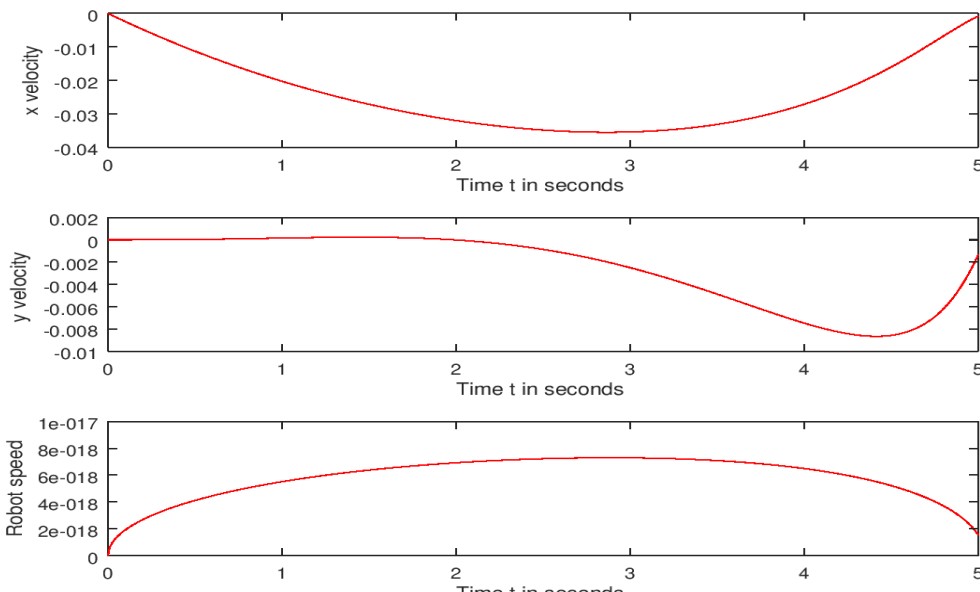

**Figure 9.** Feasible velocities.

**Case 2:** The initial position of the robot is $(x(t = 0), y(t = 0)) = (0,0)$, $\theta(t = 0) = \frac{\pi}{2}$. The other parameters in the program remain the same. The initial condition is $z_0 = [\text{zeros}(2,1);$ $\frac{\pi}{2}; 0; 0; 0; 0; 2*\text{ones}(7,1)]$, in which $\left[\text{zeros}(2,1); \frac{\pi}{2}; 0; 0; 0; 0\right]$ is the initial state vector and $2*\text{ones}(7,1)$ is the costate vector. Such graphs enable me to show the performance of the vehicle. Unlike Figure 2, which considers the starting point of the robot's trajectory at $(0,0)$ with an angle of zero radian, Figure 10 considers the same starting point $(0,0)$ but with an angle of $\frac{\pi}{2}$ with respect to the $x$ axis. In other words, we have $(x(t = 0), y(t = 0)) = (0,0)$, $\theta(t = 0) = \frac{\pi}{2}$. One can notice that the robot travels in the fourth quadrant.

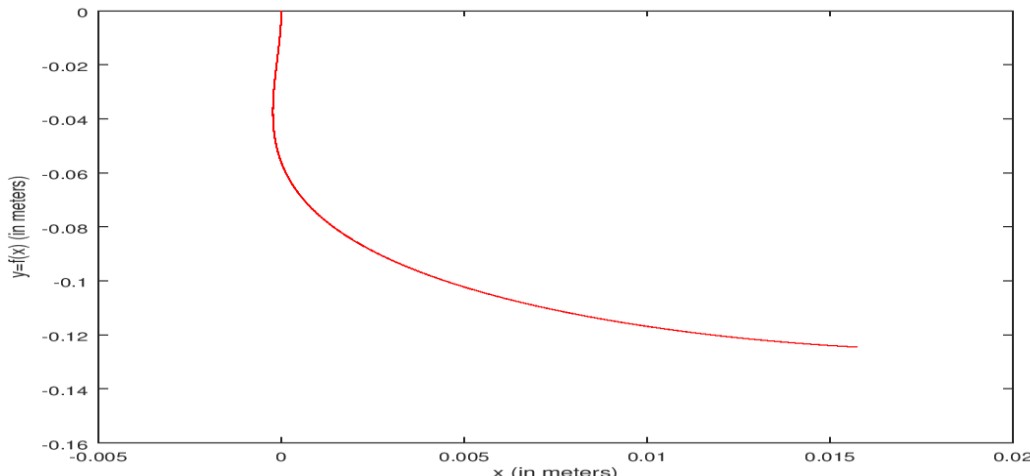

**Figure 10.** Feasible Vehicle Robot Trajectory.

Unlike Figure 4, and Figure 11 presents the first three state functions $x(t)$, $y(t)$ and $\theta(t)$ defined, respectively, by state 1, state 2 and state 3. The points $(x(t), y(t))$ define the above feasible trajectory of the robot. In this case, we have $(x(t = 0), y(t = 0)) = (0,0)$, $\theta(t = 0) = \frac{\pi}{2}$.

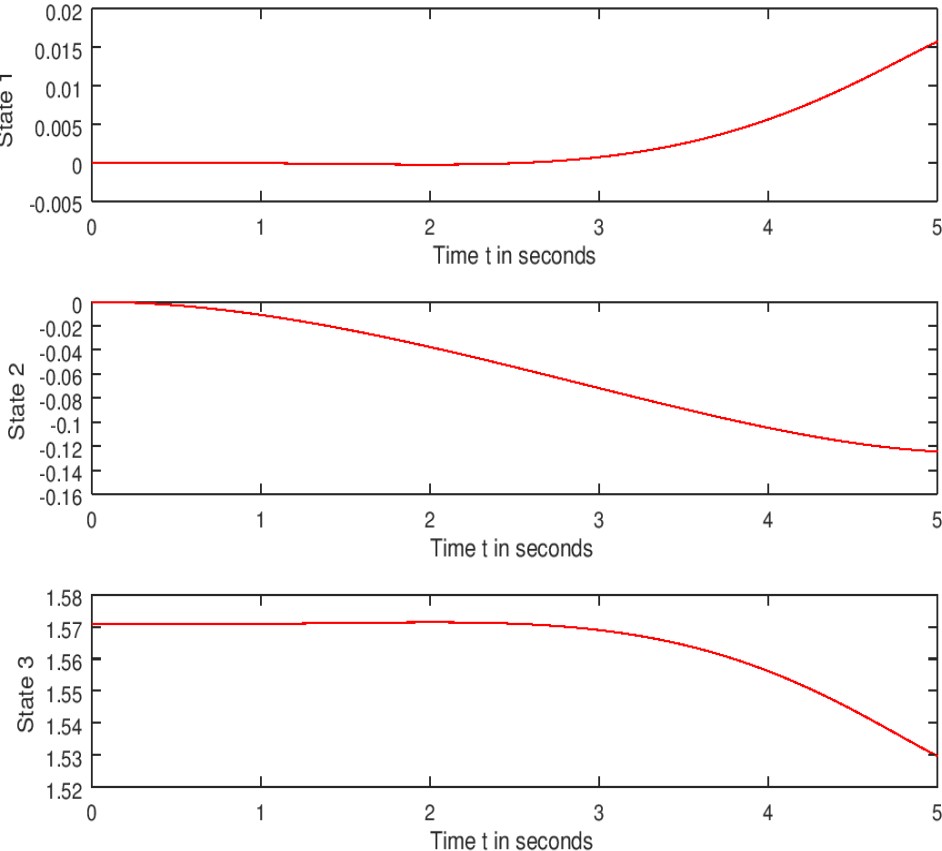

**Figure 11.** Feasible first three state trajectories.

Unlike Figure 5, Figure 12 presents the last four state functions $\delta(t)$, $\beta(t)$, $\omega(t)$ and $\varphi(t)$ given in the graph by State 1 (the steering angle), State 2 (the slip angle), State 3 (the steering angular velocity) and State4 (the heading angular velocity), respectively. Notice that state 1, state 2 and state 4 are increasing.

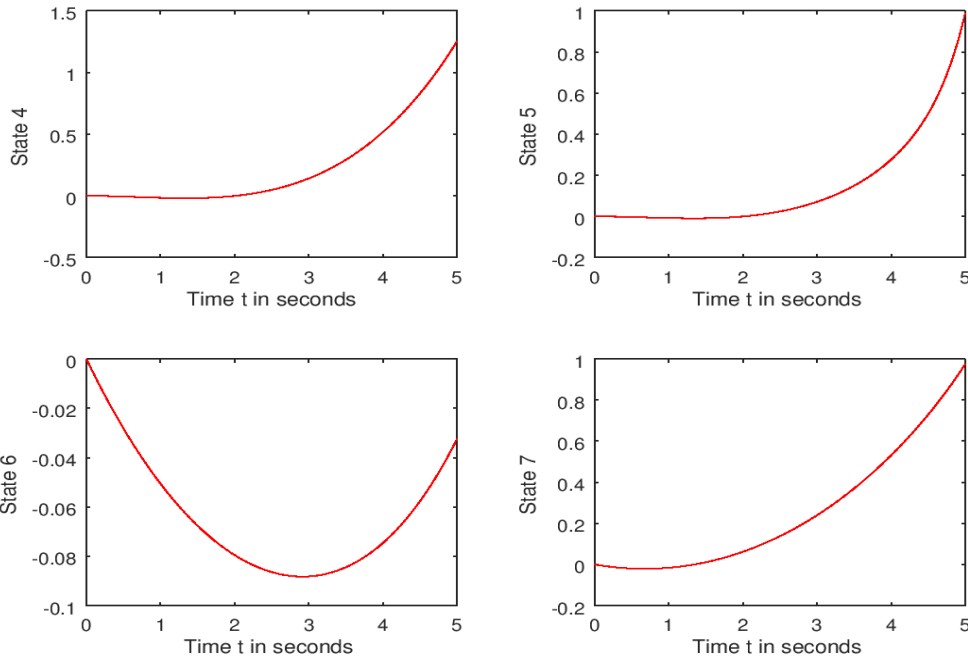

**Figure 12.** Feasible last four state trajectories: state 4, state 5, state 6, state 7.

$(x(t=0), y(t=0)) = (0,0), \theta(t=0) = \frac{\pi}{2}.$

Unlike Figure 6, Figure 13 presents the first three costate functions $\alpha_1(t)$, $\alpha_2(t)$ and $\alpha_3(t)$ (adjoint functions to $x(t)$, $y(t)$ and $\theta(t)$, respectively) given, respectively, by Costate 1, Costate 2 and Costate 3. One can notice that the first two costate functions are constant functions due to the values of their time derivatives, which are zero, given by the Equations (41) and (42):

$$(x(t=0),\ y(t=0)) = (0,0),\ \theta(t=0) = \frac{\pi}{2}$$

**Figure 13.** Feasible first three costate trajectories.

Unlike Figure 7, Figure 14 presents the last four costate functions $\alpha_4(t)$, $\alpha_5(t)$, $\alpha_6(t)$ and $\alpha_7(t)$ (adjoint functions to $\delta(t)$, $\beta(t)$, $\omega(t)$ and $\varphi(t)$, respectively) given, respectively, by Costate 4, Costate 5, Costate 6 and Costate 7.

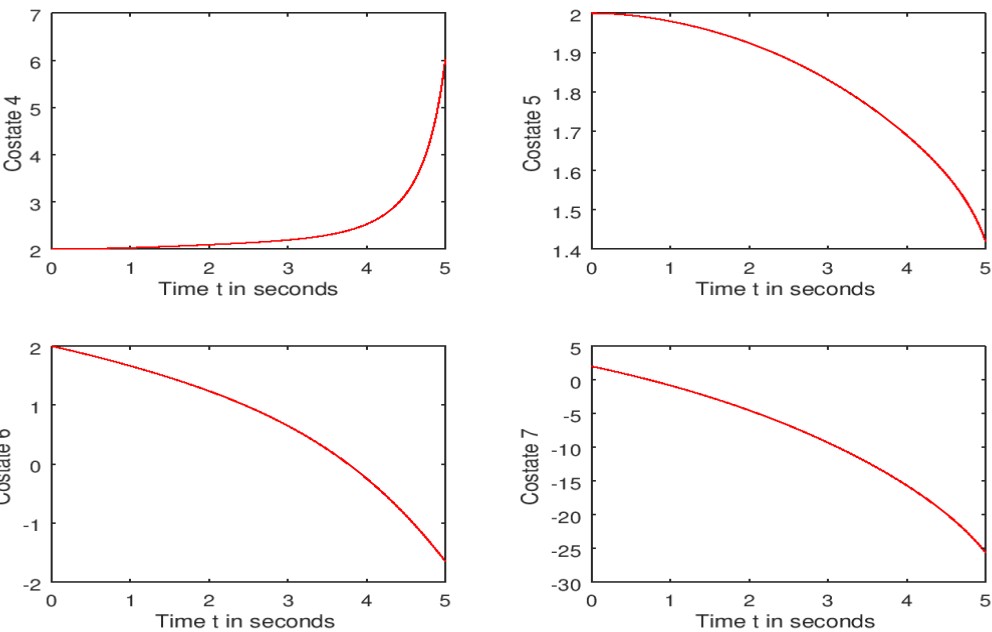

**Figure 14.** Feasible last four costate trajectories.

Unlike Figure 8, Figure 15 presents the feasible controls. One can notice that the two control strategy functions are increasing in the given interval. Control 1 and Control 2 are based on the bicycle wheels' heading angular velocity and slip angular velocity, respectively.

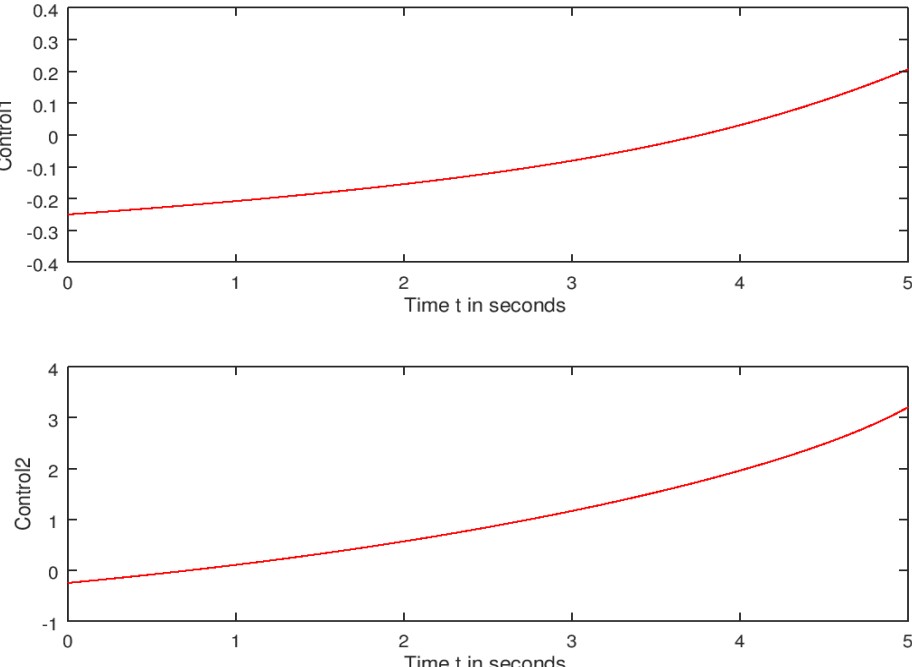

**Figure 15.** Feasible control strategies.

Figures 9 and 16 present the velocity along the $x$ direction, the velocity along the $y$ direction and the feasible speed. One can notice that the speed of the robot is obtained by combining the $x$ velocity and the $y$ velocity.

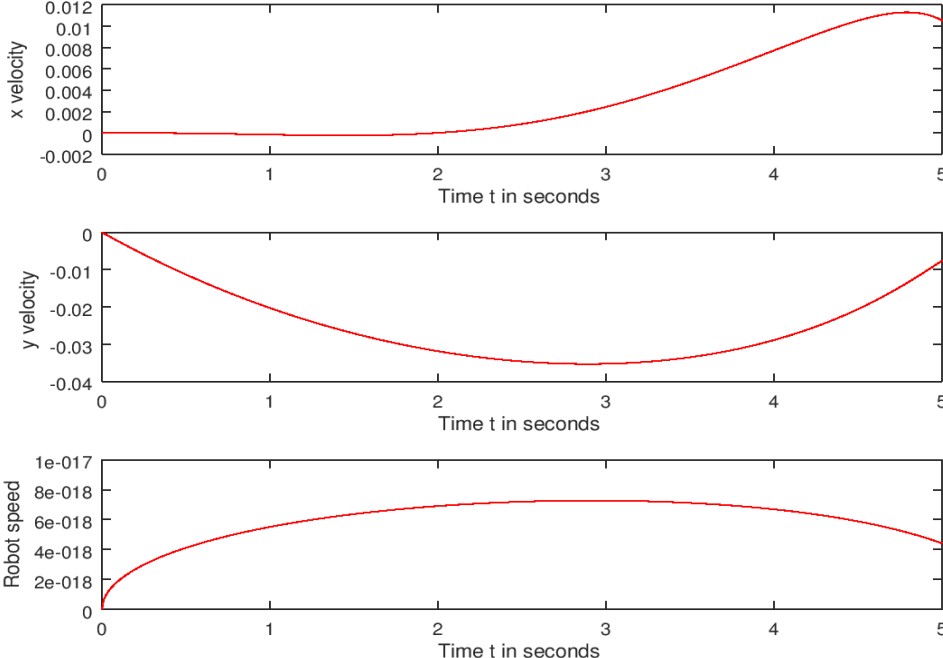

**Figure 16.** Feasible velocities.

## 6. Conclusions

The aim of this paper was to model and to control the kinematics of an autonomous bicycle robot (where the reference point is the center of gravity). The autonomous vehicle robot had to drive from a given initial state to a final state such that the running cost was minimized. Pontryagin's Minimum Principle was applied, and it derived the optimality conditions and the costate system of ordinary differential equations. The state and the costate system of ordinary differential equations were combined into a single system of ordinary differential equations. After having designed relevant initial conditions, such a system was solved and gave the feasible control strategies and the feasible state functions, from which the feasible trajectory of the robot was derived. Finally, the feasible costate functions were obtained. The obtained results enable the prediction of the performance of the autonomous bicycle robot so that it can be controlled efficiently. The designed control strategy function takes into account any disturbance of the system. Computational simulations are developed and provided to show the effectiveness of the results. One can notice from the plot of the feasible trajectory that the length of the trajectory is proportional to the magnitude of the initial costate vector provided that there is no costate component equal to zero. Future Research Directions will address the following topics: the effect of the size of the initial costate vector on the length of the robot's trajectory; the effect of the initial heading angle on the instantaneous direction of the trajectory; the path following feedback control of autonomous bicycle robots; from an optimal control problem of autonomous bicycle robots to nonlinear constrained optimization problems and their applications to robotics; the design of fifth degree Lagrange interpolating polynomials to approximate optimal control functions and their associated state functions; The optimal control of quantum systems; control of chemical processes; forensic detection of deep fake videos; data science methods to transform economic and financial data into critical information; data science methods to transform medical and biological data into critical information; the control of electromagnetic systems; the control of biological systems; machine learning applications in autonomous vehicle robots; robot path planning; free-obstacle path planning using fuzzy logic; natural language processing using linear and/or multilinear algebra; blockchain modeling and computer simulations; image analysis using differential equations; the modeling and control of photonic systems; collision-free control of an autonomous bicycle robot using a tracking policy; path planning using reinforcement learning; material science and principal component analysis; crime investigation using principal component analysis, etc.

**Funding:** This research received no external funding.

**Institutional Review Board Statement:** Not applicable.

**Informed Consent Statement:** Not applicable.

**Data Availability Statement:** Not applicable.

**Conflicts of Interest:** The authors declare no conflict of interest.

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
