# Peer review of "Control and Trajectory Planning of an Autonomous Bicycle Robot"

_computation, doi:10.3390/computation10110194_

Round 1

Reviewer 1 Report

This paper considers the modeling and the control of an autonomous bicycle robot where the reference point is the center of gravity. The controls are based on the wheels heading angular velocity and the steering angular velocity, which are developed to minimize the total running cost. A control-free state-costate system of ordinary differential equations is obtained based on the control system Hamiltonian and Pontryagin's Minimum Principle. But this paper seems to be written in a hurry, there are many problems. The detailed comments are given as follows:

1. The introduction should highlight the advantages of each contribution point by comparing it with the existing article.

2. In the bicycle model established in this paper, from formula (2) to formula (4), why the speed of the model reference center is equal to the tire rotation speed?

3. How to explain formula (7) and formula (8)? What is their physical meaning?

4. In Pontryagin's Minimum Principle, formula (27) seems to be a mistake. 

5. The paper spends a lot of space on the process of simulation, which can be simplified.

6. The constants defined in the model, such as $R$, $L$, $l_r$, etc., should be given specific values in the simulation.

7. In the simulation part, there is no clear explanation of what does each curve represent in simulation diagrams.

For the above problems, this paper should be reviewed again after being revised to meet the requirements of publication.

Author Response

Dear Sir,

I would like to inform you that I have uploaded the response to your comments and suggestions.

Reviewer 2 Report

Figures 1 and 2 are of low quality. Figures 3+ are not interpreted, i.e. missing results discussion. The introduction first half seems unrelated to the topic of bicycle robot control. Do not show your MATLAB code, the equations take care of the details. Expand the discussion and novelty to cover more than 11 citations. Perhaps you can introduce some disturbance into the system to assess the controller's robustness. 

Author Response

Dear Sir

Good day,

Please find attached the responses to your comments and suggestions

Kind Regards 

Reviewer 3 Report

The presented paper presents the control strategy of an autonomous bicycle.

I strongly suggest heavy modifications of the paper as follows:

- I suggest the author use the MDPI template. Following MDPI requirements, the presentation may improve in clarity.

- the introduction section requires profound changes: the author has cited 10 papers as a single block, without explaining why they are relevant to the research. No citations are given for other parts of the text (e.g., "Reports made by some country transport departments have shown ...", "In the literature, I discovered a set of rich and beautiful mathematical models"). This also applies in other parts of the paper, like at the beginning fo Section 4 ("It is applied in Finance, Engineering...").

- the contribution of the paper is unclear: are there in literature any papers about bicycle control? If yes, what is the difference between your approach with other methods? If not, this lack has to be highlighted.

- many figures are not cited in the text

- many parts of the model section require a wider explanation. E.g., what are a1 and a2 of equations 7 and 8? Why "optimal control theory emerges as a relevant approach to solve the problem"?

- Figure 2 must improve clarity: what is S? what is the square with the cross inside? why (xc,yc) is floating in the air?

- In Equation 28 there is a typo of ",t" between "u(t)" and "alpha*(t)"

- Equations 28 and 30 are identical

- At the end of Page 6 there is an "etc.", what does it stand for?

- The author should explain better how equations 27-33 are obtained

- Copying Matlab code within the paper is rather unusual, and in this case, it increases the length of the paper without any need. In fact, the code simply implements the equations of the model and plots the results.

- The simulation section only shows the results, without any explanation of how the two cases have been chosen (Case 2 looks like Case 1 turned 90° CCW), nor any discussion on the results.

- Has the control strategy been validated? If it is the case, a comparison between simulation and experimental results must be presented. If it is not the case, how can the reader be sure of the validity of the proposed algorithm?

Author Response

Dear Sir,

Good day,

Please find attached my responses to your comments and suggestions.

Kind Regards 

Round 2

Reviewer 1 Report

There is no comment for this time.

Author Response

There was no comments from this reviewer

Reviewer 2 Report

Figure 1 is of low quality

Figure 1 should be introduced before equation2

The text must be edited to resolve the numerous grammatical error

The Conclusion section is just a summary of the manuscript.

Author Response

Dear Sir/ Madam

Please find my reply as an attachment.

Best Regards

Dr Masiala Mavungu 

Reviewer 3 Report

The revised paper has failed to cope with some of my indications (see the list below for some examples). As a result, I think that the manuscript requires further work before the final publication.

The remarks:

- some figures are not cited in the text (e.g., Figure 1)

- many parts of the model section require a wider explanation. E.g., what are a1 and a2 of equations 7 and 8?

- Figure 1 must improve clarity: what is S? what is the square with the cross inside? why (xc,yc) is floating in the air?

- Equations 28 and 30 are identical, why?

- At the end of Page 6 there is an "etc.", what does it stand for?

- The author should explain better how equations 27-33 are obtained

- Copying Matlab code within the paper is rather unusual, and in this case, it increases the length of the paper without any need. In fact, the code simply implements the equations of the model and plots the results.

- The simulation section only shows the results, without any explanation of how the two cases have been chosen (Case 2 looks like Case 1 turned 90° CCW), nor any discussion on the results.

- Has the control strategy been validated? If it is the case, a comparison between simulation and experimental results must be presented. If it is not the case, how can the reader be sure of the validity of the proposed algorithm?

Author Response

Dear Sir/ Madam

Good day,

Please find attached my responses to your comments and suggestions.

Best Regards

Dr Masiala Mavungu 

Round 3

Reviewer 3 Report

To me Figure 1 is still of bad quality, but the rest of the paper is suitable for publishing.